# A Novel Approach for Predicting the Survival of Colorectal Cancer Patients Using Machine Learning Techniques and Advanced Parameter Optimization Methods

**DOI:** 10.3390/cancers16183205

**Published:** 2024-09-20

**Authors:** Andrzej Woźniacki, Wojciech Książek, Patrycja Mrowczyk

**Affiliations:** 1Department of Computer Science, Faculty of Computer Science and Telecommunications, Cracow University of Technology, Warszawska 24, 31-155 Cracow, Poland; andrzej.wozniackii@gmail.com; 2Oncology Clinical Department, The University Hospital in Cracow, Kopernika 50, 31-501 Cracow, Poland; pmrowczyk@su.krakow.pl

**Keywords:** colorectal cancer, survival prediction, parameter optimization, Optuna, RayTune, HyperOpt

## Abstract

**Simple Summary:**

Colorectal cancer remains a major health challenge with high mortality rates and increasing diagnoses among younger adults. This study introduces a new method to predict patient survival using machine learning, a technique that can greatly enhance early diagnosis and treatment. The aim of our study is to apply eight different machine learning algorithms to a large data set of patients with colorectal cancer in Brazil and optimize these models with advanced parameter tuning tools. The best performing models achieved around 80% accuracy in predicting survival rates over one, three, and five years, as well as overall and cancer-specific mortality. This approach promises to improve clinical decision making by providing more accurate survival predictions, ultimately helping better patient care and management.

**Abstract:**

Background: Colorectal cancer is one of the most prevalent forms of cancer and is associated with a high mortality rate. Additionally, an increasing number of adults under 50 are being diagnosed with the disease. This underscores the importance of leveraging modern technologies, such as artificial intelligence, for early diagnosis and treatment support. Methods: Eight classifiers were utilized in this research: Random Forest, XGBoost, CatBoost, LightGBM, Gradient Boosting, Extra Trees, the k-nearest neighbor algorithm (KNN), and decision trees. These algorithms were optimized using the frameworks Optuna, RayTune, and HyperOpt. This study was conducted on a public dataset from Brazil, containing information on tens of thousands of patients. Results: The models developed in this study demonstrated high classification accuracy in predicting one-, three-, and five-year survival, as well as overall mortality and cancer-specific mortality. The CatBoost, LightGBM, Gradient Boosting, and Random Forest classifiers delivered the best performance, achieving an accuracy of approximately 80% across all the evaluated tasks. Conclusions: This research enabled the development of effective classification models that can be applied in clinical practice.

## 1. Introduction

Cancer is currently one of the leading causes of death worldwide, and colorectal cancer is one of the most significant due to its high incidence and mortality rates. In 2020, nearly 2 million new cases of colorectal cancer were diagnosed, and close to a million people died from the disease [1]. Table 1 provides detailed statistics on morbidity and mortality rates by continent.

The highest number of cases has been detected in China, the USA, and Japan [2]. According to the International Agency for Research on Cancer, the number of cases is projected to increase to 3.2 million by 2040, resulting in 1.6 million deaths. These figures represent increases of 63% and 73.4%, respectively, compared with 2020 [3]. There is also a concerning trend in the rising incidence of this type of cancer among young people. Studies conducted in the USA showed that between 2004 and 2016, the percentage increase in incidence was 7.9% for the 20–29 age group, 4.9% for the 30–39 age group, and 1.6% for the 40–49 age group. An analysis of data from 1975 to 2010 estimates a 90% increase in the incidence of colorectal cancer in the 20–30 age group by 2030 [4]. According to research [5], by 2030, 15% of colorectal cancer cases will be diagnosed in young people. Unfortunately, the causes of early onset are still unknown. Participants in studies often lack typical risk factors, such as a family history of this type of cancer or the presence of polyps. However, colorectal cancer is still predominantly diagnosed in older individuals, with the average age of diagnosis being 68 for men and 72 for women. Potential risk factors include having first-degree relatives with colorectal cancer and lifestyle factors such as obesity, physical inactivity, and alcohol consumption [6]. The main treatment methods include surgical interventions, chemotherapy, radiotherapy, and immunotherapy [7].

The rising number of cancer cases is placing increasing pressure on healthcare systems worldwide. To address this challenge, modern technologies, particularly artificial intelligence, are becoming essential for supporting the diagnosis and treatment of patients. Machine learning methods have already been successfully applied in the diagnosis of various cancers, including breast cancer [8], lung cancer [9], and liver cancer [10]. Numerous scientific studies have also highlighted the potential of machine learning in the detection of colorectal cancer, with a particular focus on deep learning techniques [11].

For example, in the research from 2024 [12], convolutional neural networks (CNNs) and ranking methods were employed to diagnose colorectal cancer using histopathological data. The authors first converted the images to grayscale and applied adaptive Gaussian filtering, followed by Otsu thresholding. An 18-layer convolutional network, combined with a ranking method, was used for classification, achieving an accuracy of 91%, precision of 92%, and recall of 93%. Another approach [13] explored multispectral classification of tissues from colorectal cancer using CNN. This task involved a three-class classification: benign hyperplasia (BH), intraepithelial neoplasia (IN), and carcinoma (Ca). The research also emphasized image segmentation, achieving a classification accuracy of 99.17%, with a Jaccard similarity coefficient (JSC) of 0.86 and a dice similarity coefficient (DSC) of 0.90 for segmentation. Machine learning methods have also been applied beyond colorectal cancer diagnosis, such as in predicting mutation signatures from histopathological images and forecasting cancer recurrence [14]. In this work, the authors combined CNNs with support vector machines (SVMs). The models designed for predicting cancer recurrence achieved AUC values ranging from 0.63 to 0.93 on the test set, with similar results on the training set. This consistency indicates that the model is not overfitted and has strong generalization capabilities.

In recent years, machine learning methods have gained significant traction in the field of oncology, particularly in predicting patient survival times. This task is crucial, as it holds immense value for patients, clinicians, researchers, and decision-makers alike. Predicting life expectancy is fundamental to many clinical decisions in oncology, such as determining the likelihood of cancer recurrence or patient mortality. Traditionally, these estimates are based on the cancer’s location and stage, but the complexity of this task often involves numerous variables that surpass the analytical capabilities of a physician during a patient consultation. Advanced machine learning techniques address this challenge by analyzing complex relationships between variables, enabling more accurate life expectancy predictions [15]. For instance, in a separate investigation [16], researchers predicted the survival period of breast cancer patients using the NIH SEER dataset, which includes data from 2024 patients. The work achieved 72% accuracy in predicting two-year survival using a random forest algorithm. Comparable efforts have also been conducted for lung cancer patients [17], utilizing 12 demographic and clinical features. The best results were obtained using an 11-layer deep neural network, with a classification accuracy of 88.58%. However, when it comes to predicting the survival of patients with colorectal cancer, there is a scarcity of research in the literature. One notable exception is the recent effort in [18], where the authors conducted five types of classification: predicting one-year, three-year, and five-year survival, as well as determining whether the patient died and whether the death was cancer-related. The study used publicly available data from a hospital in São Paulo, covering the years 2020–2021. The XGBoost algorithm achieved over 74% accuracy across all five classification tasks, with AUC values exceeding 82%. These results demonstrate the feasibility of building effective models to predict survival in patients with colorectal cancer and highlight the need for further research to improve model efficiency and generalizability.

Building on this foundation, the authors of this article aimed to develop more effective classification models for predicting the survival of colorectal cancer patients. They used an expanded dataset from São Paulo, covering the years 2000–2023 (including additional data from 2021 to 2022). The primary innovation in this work is the use of a broader range of classifiers and the application of advanced libraries to optimize classifier parameters, such as Optuna, RayTune, and HyperOpt. Proper selection of classifier parameters is crucial for achieving high classification accuracy on test sets. Additionally, our research emphasizes model explainability, which is a critical aspect of medical data analysis.

## 2. Materials And Methods

### 2.1. Dataset

The research utilized a substantial dataset comprising information on 72,961 colorectal cancer patients. This publicly available dataset, sourced from the Hospital-Based Cancer Registries of São Paulo and coordinated by the Fundação Oncocentro de São Paulo [19], spans the years 2000 to 2023. The extensive size of this dataset enables the development of robust classification models for predicting patient survival. It includes comprehensive data on both clinical and demographic factors.

Figure 1 illustrates the distribution of patients by age, highlighting that the disease predominantly affects the elderly. However, the percentage of patients aged 0–49 is 22.41%.

Figure 2 shows a survival curve indicating the percentage of survival over time in months. The survival rates at one year, three years, and five years are highlighted on the graph, along with specific survival percentages at those time points. At one year, the survival rate is around 77%, at three years, it drops to 60%, and at five years, the rate is approximately 54%.

The initial dataset consisted of 107 columns. Due to issues such as incomplete data, inadequate descriptions, or the irrelevance of certain features for survival prediction, 49 columns were removed. This resulted in a final dataset with 58 remaining feature columns. Table 2 details these features. The Appendix A include an appendix with comprehensive information on all features in the dataset.

Columns with text data were encoded using label encoding. The following columns contained missing values, which were addressed using the nearest neighbor algorithm with the specified parameters. Before being input into the classifiers, the data were normalized using the standardized normal distribution.

Consistent with the approach of the study [18], several classification problems were explored based on this dataset. For this purpose, the following columns were prepared:overall_death: A binary variable where a value of 1 indicates that the patient has died, and a value of 0 signifies that the patient is either still alive or under follow-up.cancer_death: A binary variable similar to “overall_death” but specifically for cancer-related deaths. A value of 1 indicates that the patient died from cancer, while a value of 0 indicates that the patient died from other causes or is still alive.alive_year1, alive_year3, alive_year5: Binary variables indicating whether the patient is still alive 1, 3, or 5 years after diagnosis. A value of 1 signifies that the patient is still alive, and a value of 0 indicates that the patient has died or is no longer under follow-up after the specified period.

During the experiments, the data were split into a training set comprising 75% of the data and a test set comprising the remaining 25%, which were used for the final evaluation of the model.

Table 3 displays the number of samples in each training and test set for each classification problem.

### 2.2. Machine Learning Methods

In the conducted experiments, several widely used and highly effective classification methods were employed, including Random Forest [20] XGBoost [21], CatBoost [22], LightGBM [23], Gradient Boosting [24], Extra Trees [25], the k-nearest neighbor algorithm (KNN) [26], and decision trees [27]. These methods were optimized using various frameworks designed for tuning machine learning model parameters. The following libraries were utilized:Optuna [28] is a hyperparameter optimization framework that allows for automated and efficient searching of the best parameters for machine learning models. It utilizes a process known as “define-by-run”, where the search space is dynamically constructed during execution, offering greater flexibility compared with static hyperparameter configurations. Optuna integrates advanced techniques such as Bayesian optimization and early stopping to efficiently explore the parameter space and reduce training time. Moreover, it supports both single-objective and multiobjective optimization, making it a powerful tool for complex machine learning tasks. Its simplicity and performance have made it increasingly popular for tuning machine learning models.HyperOpt [29] is a widely adopted library designed for hyperparameter optimization that leverages distributed asynchronous search algorithms. It provides support for both Bayesian optimization and Tree-structured Parzen Estimator (TPE), which helps to efficiently explore hyperparameter spaces. The framework can be easily adapted to various machine learning tasks, including deep learning, and supports parallel computation, making it highly scalable. Its flexibility allows users to optimize complex objective functions, including loss functions in deep neural networks.RayTune [30] is a scalable hyperparameter tuning library that integrates seamlessly with the Ray distributed computing framework. Designed for large-scale parallel hyperparameter optimization, RayTune supports advanced search algorithms, such as population-based training (PBT), asynchronous hyperband, and Bayesian optimization. Its ability to scale effortlessly across multiple GPUs or clusters makes it highly suitable for tuning complex models, including those in deep learning or reinforcement learning. RayTune also features easy integration with other popular machine learning libraries, allowing users to conduct large experiments with minimal setup.

In recent years, the three frameworks mentioned above have gained popularity for their ability to often outperform traditional optimization methods, such as grid search or random search, and frequently surpass the results of evolutionary methods like genetic algorithms. A primary objective of this work is to compare these three modern tools in the context of predicting patient survival with colorectal cancer.

Table 4 displays the parameters that were optimized for each classifier. It is evident that classifiers within the gradient-boosting category have a particularly large number of parameters, making parameter optimization techniques essential for selecting the appropriate values. Proper parameter selection is a crucial step in developing effective machine learning models.

### 2.3. Metrics

In this research, standard metrics derived from the confusion matrix were used and calculated on the test set. Additionally, ROC curves were generated. The formulas for the selected metrics are provided below.
(1)$$Accuracy=\frac{TP+TN}{TP+TN+FP+FN}$$
(2)$$Precision=\frac{TP}{TP+FP}$$
(3)$$Recall=\frac{TP}{TP+FN}$$
(4)$$F1=\frac{2*Precision*Recall}{Precision+Recall}=\frac{2*TP}{2*TP+FP+FN}\;$$
where: *TP* is the number of True Positives, *TN* is the number of True Negatives, *FP* is the number of False Positives, *FN* is the number of False Negatives.

### 2.4. Experiment Schema

Figure 3 illustrates the experimental design.

The experiments were carried out using a dataset from the Hospital-Based Cancer Registries of São Paulo, coordinated by the Fundação Oncocentro de São Paulo. Initial preprocessing involved removing unnecessary columns, standardizing the data, and encoding categorical columns using naive methods. The data were then divided into a training set (75%) and a test set (25%).

Eight widely used and validated classifiers were used: Random Forest, XGBoost, CatBoost, LightGBM, Gradient Boosting, Extra Trees, KNN, and Decision Tree. A key aspect of the experiment was the optimization of parameters using three frameworks: Optuna, RayTune, and HyperOpt. Classification accuracy and F1-score were used as primary metrics, calculated in the test set. Detailed descriptions of the individual components of the experiment can be found in the Section 2.1, Section 2.2 and Section 2.3.

## 3. Results

In this section, we present the results of our research. They are organized into three subsections, each corresponding to one of the parameter selection methods used. For each optimization method, six classifiers were optimized, and the following metrics were calculated in the test set: precision, F1-score, precision, and recall. To ensure robust performance evaluation, 10-fold cross-validation was applied during training, using the stratified version of K-Folds to maintain balanced class distributions across the folds. For the top-performing model within each optimization method, additional analyses including the confusion matrix and ROC curve were generated.

The experiments were conducted on a computer with the following specifications.

AMD Radeon RX 6600.

AMD Ryzen 5 5600G.

32 GB RAM.

The source code was developed in Python 3.11, using libraries such as scikit-learn [31], Pandas [32], and NumPy [33]. Detailed results, including confusion matrices and ROC curves, are provided in the Appendix A because of the extensive number of experiments which could not be fully included in the main body of the article.

### 3.1. Optuna Optimization

In the first experiment, Optuna was employed to optimize the following classification models: XGBoost, LightGBM, CatBoost, Extra Trees, and KNN. The optimized parameters and their search ranges are detailed in Table 4.

Table 5 presents the results of classification models optimized using the Optuna framework. For the “alive_year1” problem, the LightGBM algorithm achieved the highest performance, with a classification accuracy of 0.8187 and an F1-score of 0.7544. In the “alive_year3” problem, the Gradient Boosting algorithm was the most effective, achieving a classification accuracy of 0.7861 and an F1-score of 0.7811. For the “alive_year5” problem, the CatBoost algorithm performed best, with a classification accuracy of 0.8185 and an F1-score of 0.7615. Regarding the “overall_death” problem, Gradient Boosting yielded the top results, with a classification accuracy of 0.7888 and an F1-score of 0.7880. In the “cancer_death” problem, LightGBM delivered the best results, with a classification accuracy of 0.7959 and an F1-score of 0.7836. Overall, most classifiers performed well, although XGBoost was the least effective, with a classification accuracy slightly above 60%. The other boosting algorithms demonstrated similar and high prediction accuracy.

### 3.2. Raytune Optimization

In the second experiment, the optimization framework was switched to the RayTune library for selecting classifier parameters. The set of classifiers and the range of parameters used remained the same as in the previous section.

Table 6 presents the results obtained using the RayTune library for model optimization. The results are comparable to those achieved with the Optuna framework and are notably high. For the “alive_year1” classification problem, the CatBoost algorithm delivered the best performance, with a classification accuracy of 0.8167 and an F1-score of 0.7511. Similarly, in the “alive_year3” problem, CatBoost again achieved the top results, with a classification accuracy of 0.7834 and an F1-score of 0.7786. In the “alive_year5” problem, LightGBM emerged as the best performer, yielding a classification accuracy of 0.8164 and an F1-score of 0.7578. For the “overall_death” problem, LightGBM also produced the best results, with a classification accuracy of 0.7879 and an F1-score of 0.7871. In the “cancer_death” problem, CatBoost proved to be the most effective, achieving a classification accuracy of 0.7955 and an F1-score of 0.7835. Overall, CatBoost was the top-performing classifier for three of the problems, while LightGBM excelled in two. The classifiers optimized using RayTune demonstrated similar classification accuracy to those optimized with Optuna.

### 3.3. Hyperopt Optimization

In the final experiment, the HyperOpt library was used for parameter optimization, while the rest of the experimental setup remained unchanged.

Table 7 presents the results obtained using the HyperOpt library for parameter optimization. This method also produced effective classification models, with an accuracy in all problems of around 80%. For the “alive_year1” problem, the CatBoost model achieved the best performance, with a classification accuracy of 0.8189 and an F1-score of 0.7543. Similarly, for the “alive_year3” problem, CatBoost was the most effective, reaching a classification accuracy of 0.7864 and an F1-score of 0.7820. In the “alive_year5” problem, the Gradient Boosting algorithm delivered the top results, with a classification accuracy of 0.8184 and an F1-score of 0.7620. For the “overall_death” problem, Gradient Boosting also excelled, achieving a classification accuracy of 0.7907 and an F1-score of 0.7897. In the “cancer_death” problem, the LightGBM classifier achieved the highest performance, with a classification accuracy of 0.7954 and an F1-score of 0.7835. Overall, ensemble methods using boosting continued to deliver the best results, while KNN and decision tree classifiers showed notably lower performance. The classification efficiency of the models optimized with HyperOpt was consistent with the results obtained with Optuna and RayTune.

### 3.4. Comparison of Results Using Optuna And Grid Search

To further validate our results, we performed additional comparisons using Grid Search optimization for the best-performing models. As shown in Table 8, the accuracy obtained with Grid Search was consistently lower compared with the accuracy after applying our chosen optimization methods, such as Optuna. This demonstrates the superior performance of more advanced adaptive methods that can better navigate the complex parameter space. We limited the Grid Search comparison to only the best classifiers, as performing a fully comprehensive comparison between random search and grid search on all classifiers would have been computationally prohibitive due to time constraints.

### 3.5. Explainability of Models

Model explainability is crucial in designing machine learning models, particularly in medical applications. To address this, the authors incorporated model interpretability into their experiments using one of the most widely recognized methods: SHAP (SHapley Additive exPlanations) values [34]. Due to the complexity of our current experiment, we chose not to perform feature selection analyses at this stage. This will be a focus of future work, where SHAP values will play a more critical role. For now, SHAP was used primarily to highlight the key features that influenced model decisions, which is important for clinical practice and the eventual deployment of algorithms in healthcare settings.

The SHAP summary plot for overall death (Figure 4) highlights the most influential features affecting the prediction of the model. Clinical staging (EC), year of diagnosis (ANODIAG), and absence of recurrence (RECENHUM) emerge as key factors, with clinical staging showing the highest impact. Surgery and the health care provider also play moderate roles in driving the model’s predictions.

Similarly, the SHAP summary plot for cancer death (Figure 5) reveals clinical staging (EC) as the dominant predictor, along with year of diagnosis (ANODIAG) and treatment-related factors such as chemotherapy (QUIMIO) and radiotherapy (RADIO). The distributions of the SHAP values provide insight into how these characteristics influence the predicted risk of cancer-specific mortality.

The SHAP summary plots for survival predictions at different time points reveal interesting patterns in the factors that drive the models. In year 1 (Figure 6), clinical staging (EC) emerges as the dominant predictor, followed by the absence of recurrence (RECENHUM) and year of diagnosis (ANODIAG). This indicates that early survival is heavily influenced by the stage of cancer and whether a recurrence has been detected.

In year 3 (Figure 7) and year 5 (Figure 8), the plot changes slightly, and the clinical staging (EC) continues to be important, but now age (IDADE) joins the year of diagnosis (ANODIAG) as a key factor. Long-term survival appears to be more closely related to the age at which the cancer was diagnosed and the age of the patient.

The analysis highlights the evolving importance of clinical staging, diagnosis year, and patient age as key predictors over time. This interpretability not only strengthens our confidence in the model but also helps clinicians focus on the most relevant factors when considering patient treatment plans and long-term prognosis.

## 4. Discussion

Colorectal cancer is one of the most common cancers, characterized by high mortality rates and an increasing incidence among young adults. To improve the diagnosis and treatment of this disease, modern technologies are essential, particularly artificial intelligence and machine learning methods. The research carried out by the authors represents one of the pioneering efforts in predicting patient survival for colorectal cancer. This study used a substantial dataset from Brazil that included patient data collected between 2000 and 2023. The extensive dataset enables reliable research and the development of classification models with high generalizability and minimal overfitting.

The authors addressed five distinct problems: one-year survival (referred to as alive_year1), three-year survival (alive_year3), five-year survival (alive_year5), overall death prediction (overall_death), and cancer-specific death prediction (cancer_death). Each of these problems was framed as a separate binary classification task. A range of classifiers known for their high performance across various problems were employed, including XgBoost, CatBoost, LightGBM, Gradient Boosting, Extra Trees, KNN, and Decision Tree. By evaluating a diverse set of models, this study aimed to identify the most effective algorithms.

The authors also conducted preliminary work with deep learning models, such as CNNs and LSTMs. However, the initial results from these models were quite poor. It is possible that the complexity of the data, considering the number of features, is not substantial enough for deep learning models to effectively learn and provide significant improvements in performance. Additionally, incorporating these models would introduce further complexity to the study without a corresponding benefit in performance.

A significant aspect of this work was the application of three popular parameter optimization frameworks: Optuna, RayTune, and HyperOpt. Despite their widespread use, there are few studies that compare these three approaches directly. Parameter optimization is a critical stage in the development of effective machine learning models, and the authors dedicated considerable effort to this aspect. This comprehensive comparison ensures that the parameters have been optimally selected, enhancing the potential effectiveness of the models in clinical practice.

Table 9 provides a comparative analysis of the best models built using these three optimization techniques for the five classification problems.

It is evident that all three parameter optimization techniques performed exceptionally well in selecting appropriate model parameters, with only minor differences among them. This shows that there are several highly effective frameworks available to optimize the parameters of the machine learning model. HyperOpt provided the best parameters for three of the problems, while Optuna excelled for two. The results of RayTune were only slightly less favorable, with differences between the frameworks being less than one percent. The best performance was achieved for predicting one-year survival, where the CatBoost classifier attained a classification accuracy of 0.8189 and an F1-score of 0.7543. This problem appears to be less complex than long-term predictions. For the three-year survival forecast, CatBoost again delivered the highest performance, with an accuracy of 0.7864 and an F1-score of 0.7820. HyperOpt was particularly effective for one-year and three-year predictions. For the prediction of five-year survival, CatBoost, optimized with Optuna, achieved the highest performance with an accuracy of 0.8185 and an F1-score of 0.7615. In predicting overall death, Gradient Boosting, optimized with HyperOpt, performed best, with an accuracy of 0.7907 and an F1-score of 0.7897. To predict cancer-specific death, the LightGBM model, optimized with Optuna, achieved the best results, with a classification efficiency of 0.7959 and an F1-score of 0.7836. In general, the models demonstrated high effectiveness in the five classification problems, with accuracies around 80%. The F1-scores were consistent with the classification efficiencies, indicating robust generalization and effective class handling. Compared with previous work [18], which also used the same dataset but only covered up to 2021 and relied on RandomForest, Naive Bayes, and XgBoost, our study represents a significant advance. We extended the dataset to include information up to 2023, explored a wider array of classifiers, and employed advanced parameter optimization frameworks. This approach led to improved results. Table 10 compares the best results from our study with those reported by Buk Cardoso et al. [18]. Furthermore, this study aligns with larger findings in the field, such as those reviewed by Kourou et al. [35], which emphasize the transformative impact of machine learning on cancer research and underscore the importance of robust and explainable models in clinical practice.

The results for each of the five problems exceed those reported in the study by [18]. According to the authors, these improvements are attributed to the use of advanced parameter optimization frameworks: Optuna, RayTune, and HyperOpt. The studies also emphasized model explainability through the use of SHAP values. Model explainability is crucial for integrating machine learning algorithms into clinical practice, as it enhances trust in these methods and encourages their adoption for diagnostic and therapeutic support across a broad spectrum of diseases. The most notable achievements of this research include the following:Developing effective machine learning models for survival prediction.Addressing five classification problems with eight different classifiers and three parameter optimization methods.Comparison of three advanced parameter optimization frameworks to determine the most effective approach.

Regarding the limitations of the research, it is important to note the following:Data Scope: The dataset is limited to patients from Brazil, which may impact the model’s generalizability to other populations. Cancer outcomes can vary significantly between countries due to factors such as diet, genetic predispositions, and differences in healthcare systems. Therefore, it is essential to validate the model on datasets from different regions to ensure it can be effectively applied across diverse populations. At present, data from other countries are not readily available, but with the growing use of machine learning and big data in healthcare, we anticipate that such data will become more accessible in the near future. This would allow for cross-country comparisons and further refinement of the model to mitigate potential biases and enhance its robustness in different contexts.Feature Selection: This study did not address feature selection. Future research should incorporate feature selection methods, such as Principal Component Analysis (PCA) or genetic algorithms, to further refine model performance.Sample Balancing: The research did not apply oversampling techniques to balance class distributions. Implementing methods like SMOTE (Synthetic Minority Oversampling Technique) in the training set could potentially improve model effectiveness.

This study has led to the development of effective classification models to predict the survival of patients with colorectal cancer, offering promising prospects for clinical application. Future research will focus on feature selection using algorithms such as Principal Component Analysis (PCA) and genetic algorithms, areas in which the authors have extensive experience [36]. Given the large number of features involved in the current study, this is a complex and multifaceted challenge. A comprehensive exploration of feature selection methods is necessary to build the most accurate and efficient models, ensuring that all relevant dimensions of the data are thoroughly analyzed and optimized. Additionally, genetic algorithms will be explored for parameter optimization, allowing for comparative studies of Optuna, RayTune, and HyperOpt frameworks alongside evolutionary methods. The use of oversampling techniques, particularly SMOTE (Synthetic Minority Oversampling Technique), will also be considered to balance sample distributions across classes, a strategy known to enhance model performance [37]. Furthermore, the research will explore other ensemble model building techniques, such as model contamination [38].

## 5. Conclusions

In this study, several effective classification models were developed to predict 1-year, 3-year and 5-year survival, as well as to predict patient death and cancer-specific mortality. These models were designed using advanced machine learning parameter optimization frameworks: Optuna, RayTune, and HyperOpt. The results achieved demonstrate promising potential for applying these models in clinical practice. However, more research is required to enhance the effectiveness of these methods and ensure their optimal performance.

## Figures and Tables

**Figure 1 cancers-16-03205-f001:**
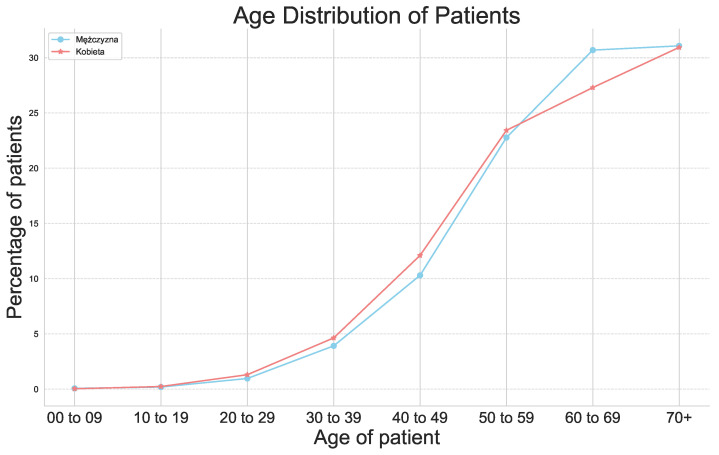
Age distribution of patients.

**Figure 2 cancers-16-03205-f002:**
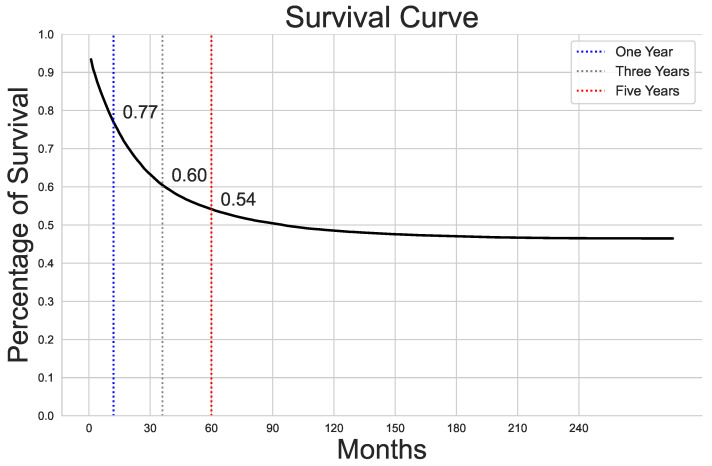
Survival curve.

**Figure 3 cancers-16-03205-f003:**
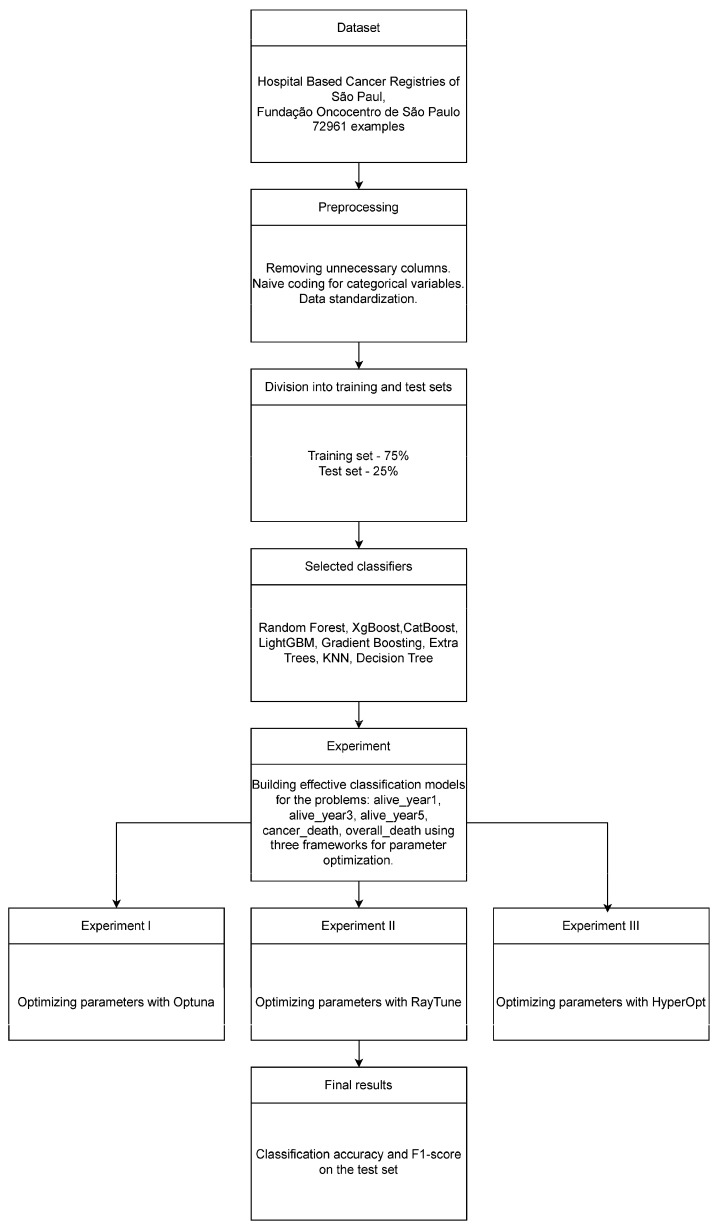
Experiment schema.

**Figure 4 cancers-16-03205-f004:**
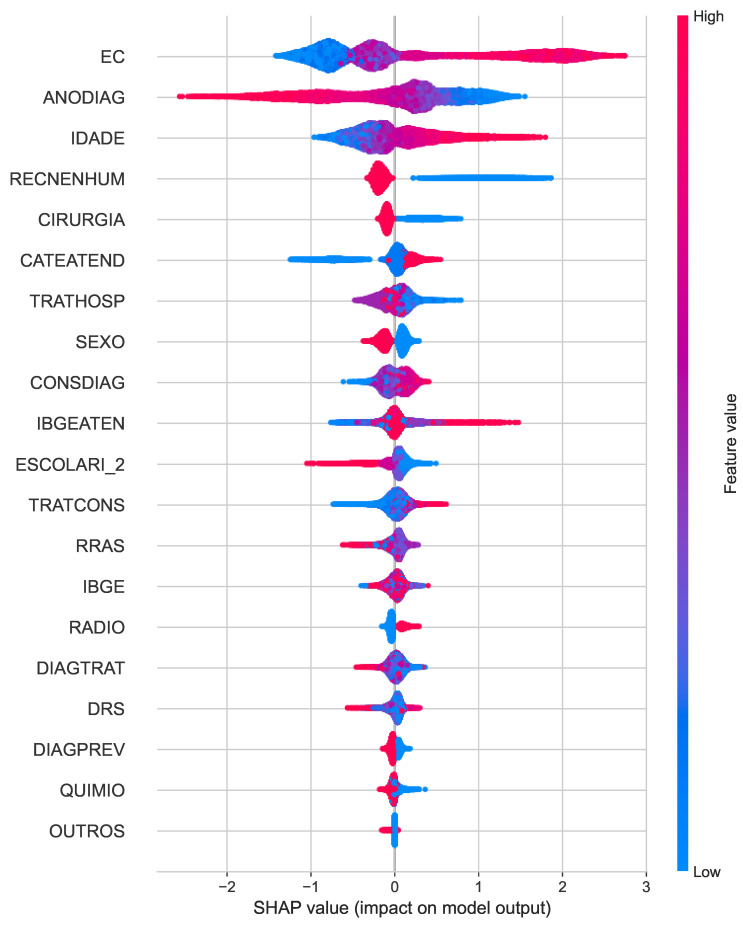
SHAP summary plots for the best classifiers of overall death.

**Figure 5 cancers-16-03205-f005:**
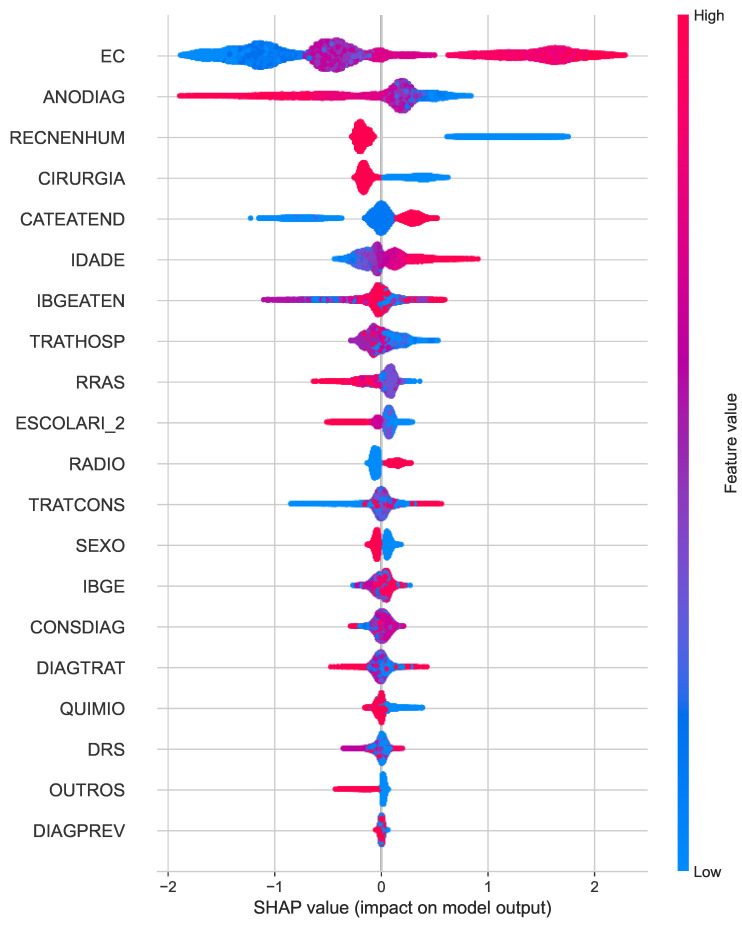
SHAP summary plots for the best classifiers of cancer death.

**Figure 6 cancers-16-03205-f006:**
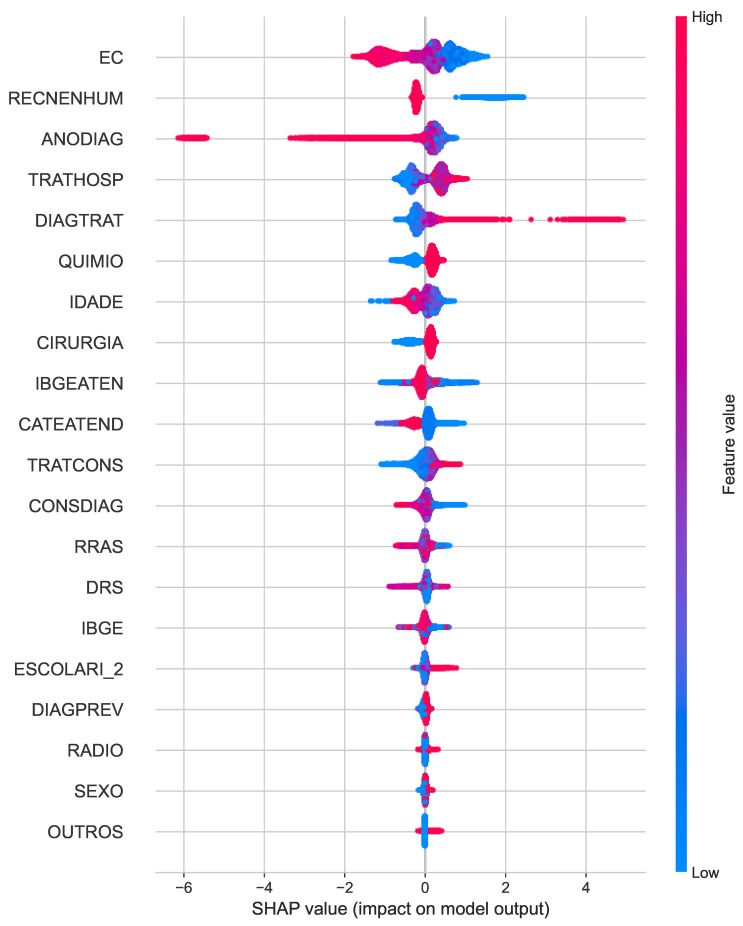
SHAP summary plots for the best classifiers for Year 1.

**Figure 7 cancers-16-03205-f007:**
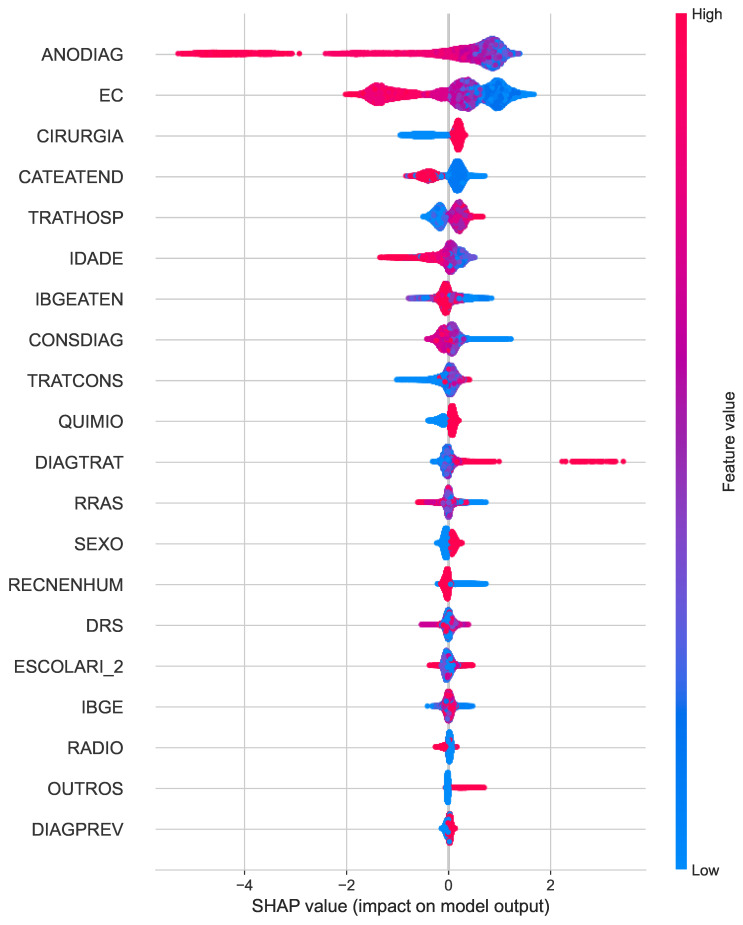
SHAP summary plots for the best classifiers for Year 3.

**Figure 8 cancers-16-03205-f008:**
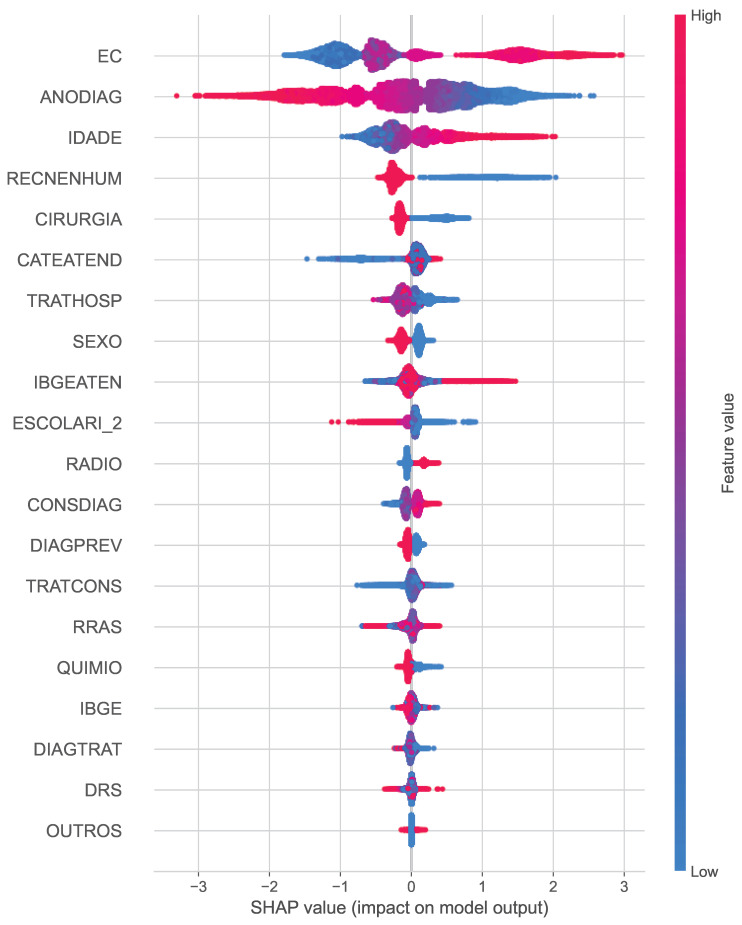
SHAP summary plots for the best classifiers for Year 5.

**Table 1 cancers-16-03205-t001:** Colorectal cancer incidence and mortality statistics [2].

Population	Number of Cases	Number of Deaths
Oceania	22,243	8036
Africa	70,428	46,087
Latin America and the Caribbean	145,120	66,155
Northern America	183,973	73,647
Europe	538,262	247,842
Asia	966,399	462,252
**Total**	**1,926,425**	**904,019**

**Table 2 cancers-16-03205-t002:** Details of dataset.

Name	Description	Data Type
SEXO	Patient’s gender	int
IDADE	Patient’s age	int
ESCOLARI	Education level	int
UFNASC	Birthplace	char
IBGE	Postal code	char
CIDADE	City of residence	char
CATEATEND	Health service provider	int
DTCONSULT	Date of first consultation	date
CLINICA	Diagnosis department	int
DIAGPREV	Diagnosis status	int
DTDIAG	Date of diagnosis	date
BASEDIAG	Basis of suspicion	int
TOPO	Cancer subgroup	char
TOPOGRUP	Cancer group	char
DESCTOPO	Cancer location	char
MORFO	Morphological code	char
DESCMORFO	Morphological subtype	char
EC	Clinical staging	char
ECGRUP	Staging group	char
T	Tumor growth	char
N	Spread to lymph nodes	char
M	Distant metastases	char
PT	PT scale	char
PN	PN scale	char
PM	PM scale	char
S	S scale	int
G	G scale	char
IDMITOTIC	Mitotic index	int
PSA	PSA classification	int
GLEASON	Gleason score	int
OUTRACLA	OUTRACLA code	char
META01	Metastasis 1	char
META02	Metastasis 2	char
META03	Metastasis 3	char
META04	Metastasis 4	char
DTTRAT	Treatment start date	date
NAOTRAT	Reason for no treatment	int
TRATAMENTO	Type of treatment	char
TRATHOSP	Treatment in hospital	char
TRATFANTES	Treatment before hospital	char
TRATFAPOS	Treatment after hospital	char
NENHUM	No treatment in hospital	boolean
CIRURGIA	Surgery in hospital	boolean
RADIOa	Radiotherapy in hospital	boolean
QUIMIO	Chemotherapy in hospital	boolean
HORMONIO	Hormone therapy in hospital	boolean
TMO	Stem cell transplant in hospital	boolean
IMUNO	Immunotherapy in hospital	boolean
OUTROS	Other methods in hospital	boolean
NENHUMANT	No treatment outside hospital	boolean
CIRURANT	Surgery outside hospital	boolean
RADIOANT	Radiotherapy outside hospital	boolean
QUIMIOANT	Chemotherapy outside hospital	boolean
HORMOANT	Hormone therapy outside hospital	boolean
TMOANT	Stem cell transplant outside hospital	boolean
IMUNOANT	Immunotherapy outside hospital	boolean
OUTROANT	Other methods outside hospital	boolean
NENHUMAPOS	No treatment anywhere	boolean
CIRURAPOS	Surgery anywhere	boolean
RADIOAPOS	Radiotherapy anywhere	boolean
QUIMIOAPOS	Chemotherapy anywhere	boolean
HORMOAPOS	Hormone therapy anywhere	boolean
TMOAPOS	Stem cell transplant anywhere	boolean
IMUNOAPOS	Immunotherapy anywhere	boolean
OUTROAPOS	Other methods anywhere	boolean
DTULTINFO	Last info date	date
ULTINFO	Last info status	int
CONSDIAG	Days from consultation to diagnosis	int
TRATCONS	Days from consultation to treatment	int
DIAGTRAT	Days from diagnosis to treatment	int
ANODIAG	Year of diagnosis	int
FAIXAETAR	Age group	char
LATERALI	Tumor laterality	char
INSTORIG	Treatment location if previously diagnosed	char
DRS	NFZ department	char
RRAS	Regional health network	char
DTPREENCH	End of treatment date	date
REGISTRADO	Registration status	char
DTRECIDIVA	Recurrence date	date
RECNENHUM	No recurrence	int
RECLOCAL	Local recurrence	int
RECREGIO	Regional recurrence	int
RECDIST	Distant metastasis	int
REC01	Metastasis type 1	char
REC02	Metastasis type 2	char
REC03	Metastasis type 3	char
REC04	Metastasis type 4	char
HABILIT2	Certification status	char

**Table 3 cancers-16-03205-t003:** Distribution of data in a dataset.

		Overall_Death	Cancer_Death	Alive_Year1	Alive_Year3	Alive_Year5
**Train**	No (Class 0)	11,256	14,426	6819	14,065	17,704
	Yes (Class 1)	12,962	9792	17,399	10,153	6514
**Test**	No (Class 0)	3752	4809	2273	4688	5902
	Yes (Class 1)	4321	3264	5800	3385	2171
**Total**	No (Class 0)	15,008	19,235	9092	18,753	23,606
**Total**	Yes (Class 1)	17,283	13,056	23,199	13,538	8685

**Table 4 cancers-16-03205-t004:** Optimized parameters for classifiers.

Name	Parameter	Range
RandomForest	n_estimators	1–200
	max_depth	2–20
	min_weight_fraction_leaf	0.0–0.5
	min_samples_split	2–10
	min_samples_leaf	1–10
	max_samples	0.7–1.0 step = 0.05
	criterion	‘gini’, ‘entropy’
XgBoost	n_estimators	1–200
	max_depth	3–10
	scale_pos_weight	1–3
	learning_rate	0.01–0.3
	subsample	0.5–1
	colsample_bytree	0.5–1
	gamma	0–1
	min_child_weight	1–10
CatBoost	iterations	100–1000
	depth	2–10
	learning_rate	0.01–0.3
	l2_leaf_reg	1 × 10^−5^ –1
	border_count	1–255
	random_strength	0–1
LightGBM	num_leaves	2–256
	max_depth	2–50
	learning_rate	0.01–0.3
	n_estimators	100–500
	min_child_samples	5–100
	subsample	0.4–1
	colsample_bytree	0.4–1
	reg_alpha	1 × 10^−8^–10.0
	reg_lambda	1 × 10^−8^–10.0
GradientBoostingClassifier	max_depth	2–50
	learning_rate	0.01–0.3
	n_estimators	100–500
	subsample	0.4–1.0
	min_samples_split	2–100
	min_samples_leaf	1–100
	max_features	[’sqrt’, ‘log2’, None]
Extra Trees	max_depth	2–50
	n_estimators	100–1000
	min_samples_split	2–100
	min_samples_leaf	1–100
	max_features	[‘sqrt’, ‘log2’, None]
Knn	n_neighbors	1–20
	weights	[‘uniform’, ‘distance’]
Decision Tree	max_depth	2–50
	min_samples_split	2–100
	min_samples_leaf	1–100
	max_features	[‘sqrt’, ‘log2’, None]

**Table 5 cancers-16-03205-t005:** Results of parameter optimization with Optuna.

		Accuracy	F1-Score	Precision	Recall	AUC
alive_year1	Random Forest	0.8126	0.7379	0.7903	0.7160	0.846
	XgBoost	0.6808	0.5707	0.5823	0.5680	0.650
	CatBoost	0.8165	0.7503	0.7885	0.7311	0.850
	LightGBM	0.8187	0.7544	0.7904	0.7356	0.855
	Gradient Boosting	0.8148	0.7503	0.7835	0.7325	0.850
	Extra Trees	0.8059	0.7328	0.7748	0.7136	0.838
	KNN	0.7848	0.7009	0.7430	0.6840	0.806
	Decision Tree	0.7907	0.7204	0.7457	0.7065	0.814
alive_year3	Random Forest	0.7803	0.7755	0.7816	0.7769	0.860
	XgBoost	0.6325	0.6303	0.6753	0.6621	0.716
	CatBoost	0.7811	0.7765	0.7753	0.7781	0.860
	LightGBM	0.7834	0.7788	0.7776	0.7806	0.865
	Gradient Boosting	0.7861	0.7811	0.7803	0.7821	0.865
	Extra Trees	0.7747	0.7671	0.7693	0.7655	0.852
	KNN	0.7360	0.7267	0.7292	0.7251	0.807
	Decision Tree	0.7516	0.7450	0.7450	0.7449	0.833
alive_year5	Random Forest	0.8133	0.7461	0.7687	0.7318	0.883
	XgBoost	0.7315	0.6615	0.6602	0.6630	0.785
	CatBoost	0.8185	0.7615	0.7716	0.7534	0.885
	LightGBM	0.8185	0.7601	0.7722	0.7508	0.888
	Gradient Boosting	0.8149	0.7570	0.7666	0.7492	0.886
	Extra Trees	0.8151	0.7576	0.7666	0.7503	0.881
	KNN	0.7786	0.6783	0.7229	0.6620	0.825
	Decision Tree	0.8012	0.7361	0.7486	0.7269	0.858
overall_death	Random Forest	0.7889	0.7746	0.7870	0.7689	0.861
	XgBoost	0.6569	0.6329	0.6755	0.6420	0.736
	CatBoost	0.7856	0.7846	0.7845	0.7848	0.869
	LightGBM	0.7868	0.7857	0.7858	0.7855	0.870
	Gradient Boosting	0.7888	0.7880	0.7877	0.7884	0.871
	Extra Trees	0.7838	0.7831	0.7828	0.7836	0.865
	KNN	0.7528	0.7513	0.7515	0.7512	0.830
	Decision Tree	0.7585	0.7574	0.7573	0.7576	0.838
cancer_death	Random Forest	0.7860	0.7849	0.7849	0.7849	0.861
	XgBoost	0.6080	0.6078	0.6260	0.6276	0.683
	CatBoost	0.7947	0.7831	0.7902	0.7789	0.864
	LightGBM	0.7959	0.7836	0.7923	0.7789	0.867
	Gradient Boosting	0.7951	0.7835	0.7906	0.7793	0.864
	Extra Trees	0.7910	0.7778	0.7880	0.7726	0.857
	KNN	0.7602	0.7456	0.7536	0.7415	0.829
	Decision Tree	0.7722	0.7575	0.7677	0.7526	0.837

Rows highlighted with a green background indicate the best performing models in terms of accuracy for the respective time frame.

**Table 6 cancers-16-03205-t006:** Results of parameter optimization with RayTune.

		Accuracy	F1-Score	Precision	Recall	AUC
alive_year1	Random Forest	0.7184	0.4181	0.3592	0.5000	0.727
	XgBoost	0.6753	0.5271	0.5490	0.5323	0.608
	CatBoost	0.8167	0.7511	0.7880	0.7322	0.847
	LightGBM	0.8165	0.7535	0.7852	0.7362	0.851
	Gradient Boosting	0.8106	0.7466	0.7754	0.7304	0.847
	Extra Trees	0.7219	0.4319	0.8293	0.5064	0.805
	KNN	0.7821	0.7058	0.7343	0.6916	0.792
	Decision Tree	0.7466	0.5823	0.7049	0.5840	0.730
alive_year3	Random Forest	0.6984	0.6707	0.6995	0.6690	0.765
	XgBoost	0.6329	0.6287	0.6890	0.6672	0.733
	CatBoost	0.7834	0.7786	0.7775	0.7800	0.859
	LightGBM	0.7827	0.7779	0.7769	0.7793	0.863
	Gradient Boosting	0.7822	0.7772	0.7764	0.7783	0.862
	Extra Trees	0.6918	0.6412	0.7212	0.6481	0.808
	KNN	0.7279	0.7187	0.7205	0.7174	0.794
	Decision Tree	0.7023	0.6882	0.6946	0.6858	0.748
alive_year5	Random Forest	0.7311	0.4223	0.3655	0.5000	0.785
	XgBoost	0.7318	0.6826	0.6751	0.6991	0.786
	CatBoost	0.8161	0.7577	0.7685	0.7493	0.878
	LightGBM	0.8164	0.7578	0.7691	0.7491	0.887
	Gradient Boosting	0.8137	0.7553	0.7650	0.7475	0.884
	Extra Trees	0.7333	0.4333	0.7772	0.5049	0.833
	KNN	0.7767	0.6911	0.7153	0.6784	0.810
	Decision Tree	0.7480	0.6109	0.6697	0.6023	0.769
overall_death	Random Forest	0.7342	0.7041	0.7391	0.6987	0.814
	XgBoost	0.6257	0.5646	0.6885	0.6023	0.699
	CatBoost	0.7860	0.7850	0.7849	0.7852	0.868
	LightGBM	0.7879	0.7871	0.7869	0.7873	0.869
	Gradient Boosting	0.7860	0.7849	0.7849	0.7850	0.869
	Extra Trees	0.7572	0.7525	0.7609	0.7512	0.835
	KNN	0.7479	0.7465	0.7467	0.7464	0.820
	Decision Tree	0.6895	0.6859	0.6883	0.6853	0.755
cancer_death	Random Forest	0.7336	0.7286	0.7359	0.7277	0.812
	XgBoost	0.4646	0.4180	0.5871	0.5385	0.704
	CatBoost	0.7955	0.7835	0.7915	0.7790	0.863
	LightGBM	0.7931	0.7813	0.7886	0.7770	0.865
	Gradient Boosting	0.7909	0.7794	0.7856	0.7756	0.861
	Extra Trees	0.7427	0.7048	0.7657	0.6998	0.836
	KNN	0.7542	0.7409	0.7461	0.7378	0.816
	Decision Tree	0.6327	0.5951	0.6124	0.5962	0.635

Rows highlighted with a green background indicate the best performing models in terms of accuracy for the respective time frame.

**Table 7 cancers-16-03205-t007:** Results of parameter optimization with HyperOpt.

		Accuracy	F1-Score	Precision	Recall	AUC
alive_year1	Random Forest	0.8123	0.7366	0.7910	0.7145	0.847
	XgBoost	0.7069	0.5463	0.5980	0.5524	0.654
	CatBoost	0.8189	0.7543	0.7912	0.7352	0.851
	LightGBM	0.8174	0.7532	0.7880	0.7348	0.854
	Gradient Boosting	0.8157	0.7519	0.7843	0.7343	0.851
	Extra Trees	0.7203	0.4255	0.8305	0.5034	0.806
	KNN	0.7848	0.7009	0.7430	0.6840	0.806
	Decision Tree	0.7541	0.6263	0.7031	0.6162	0.763
alive_year3	Random Forest	0.7784	0.7738	0.7726	0.7757	0.858
	XgBoost	0.6591	0.6586	0.6669	0.6702	0.708
	CatBoost	0.7864	0.7820	0.7808	0.7838	0.864
	LightGBM	0.7850	0.7804	0.7792	0.7820	0.866
	Gradient Boosting	0.7840	0.7791	0.7781	0.7804	0.863
	Extra Trees	0.6922	0.6432	0.7190	0.6493	0.809
	KNN	0.7360	0.7267	0.7292	0.7251	0.807
	Decision Tree	0.6888	0.6798	0.6802	0.6794	0.754
alive_year5	Random Forest	0.8139	0.7430	0.7722	0.7263	0.882
	XgBoost	0.7276	0.6636	0.6597	0.6688	0.789
	CatBoost	0.8180	0.7617	0.7706	0.7546	0.884
	LightGBM	0.8174	0.7607	0.7698	0.7533	0.888
	Gradient Boosting	0.8184	0.7620	0.7712	0.7545	0.887
	Extra Trees	0.7332	0.4315	0.8187	0.5042	0.830
	KNN	0.7786	0.6783	0.7229	0.6620	0.825
	Decision Tree	0.7587	0.6507	0.6879	0.6380	0.798
overall_death	Random Forest	0.7879	0.7733	0.7864	0.7674	0.861
	XgBoost	0.6537	0.6221	0.6829	0.6365	0.740
	CatBoost	0.7860	0.7850	0.7849	0.7852	0.869
	LightGBM	0.7899	0.7891	0.7889	0.7895	0.871
	Gradient Boosting	0.7907	0.7897	0.7896	0.7899	0.870
	Extra Trees	0.7555	0.7511	0.7584	0.7499	0.831
	KNN	0.7528	0.7513	0.7515	0.7512	0.830
	Decision Tree	0.6990	0.6986	0.6989	0.6999	0.770
cancer_death	Random Forest	0.7863	0.7854	0.7852	0.7855	0.865
	XgBoost	0.6530	0.6513	0.6581	0.6637	0.730
	CatBoost	0.7944	0.7826	0.7899	0.7783	0.864
	LightGBM	0.7954	0.7835	0.7911	0.7792	0.866
	Gradient Boosting	0.7895	0.7780	0.7842	0.7742	0.862
	Extra Trees	0.7410	0.7014	0.7664	0.6968	0.835
	KNN	0.7602	0.7456	0.7536	0.7415	0.829
	Decision Tree	0.6758	0.6302	0.6718	0.6317	0.731

Rows highlighted with a green background indicate the best performing models in terms of accuracy for the respective time frame.

**Table 8 cancers-16-03205-t008:** Comparison of optimization methods for best classifiers.

Classification Problem	Best Classifier	Accuracy	Grid Search Accuracy
alive_year1	LightGBM	0.8187	0.7235
alive_year3	Gradient Boosting	0.7861	0.7441
alive_year5	CatBoost	0.8185	0.7044
overall_death	Random Forest	0.7889	0.7122
cancer_death	LightGBM	0.7959	0.7412

**Table 9 cancers-16-03205-t009:** Summary of results for all parameter optimization methods.

		Optuna		RayTune		HyperOpt	
alive_year1	Accuracy	LightGBM	0.8187	CatBoost	0.8167	CatBoost	0.8189
	F1-Score		0.7544		0.7511		0.7543
alive_year3	Accuracy	Gradient Boosting	0.7861	CatBoost	0.7834	CatBoost	0.7864
	F1-Score		0.7811		0.7786		0.7820
alive_year5	Accuracy	CatBoost	0.8185	LightGBM	0.8164	Gradient Boosting	0.8184
	F1-Score		0.7615		0.7578		0.7620
overall_death	Accuracy	Gradient Boosting	0.7888	LightGBM	0.7879	Gradient Boosting	0.7907
	F1-Score		0.7880		0.7871		0.7897
cancer_death	Accuracy	LightGBM	0.7959	CatBoost	0.7955	LightGBM	0.7954
	F1-Score		0.7836		0.7835		0.7835

Rows highlighted with a green background indicate the best performing models in terms of accuracy for the respective time frame.

**Table 10 cancers-16-03205-t010:** Comparison of results with existing literature.

Problem	Metrics	Cardoso et al. [18]	Current Study
alive_year1	Accuracy [%]	77.4	81.89
alive_year3	Accuracy [%]	74.7	78.64
alive_year5	Accuracy [%]	77.9	81.85
overall_death	Accuracy [%]	77.7	79.07
overall_death	Accuracy [%]	77.1	79.59

## Data Availability

The data used in this research are publicly available and can be accessed at https://fosp.saude.sp.gov.br/ (accessed on 17 September 2024).

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
