# Peer review of "A Novel Approach for Predicting the Survival of Colorectal Cancer Patients Using Machine Learning Techniques and Advanced Parameter Optimization Methods"

_cancers, 2024, doi:10.3390/cancers16183205_

Round 1

Reviewer 1 Report

Comments and Suggestions for Authors

The present article deals with a particularly important topic for the precision medicine of colorectal cancer: the use of artificial intelligence for the benefit of patients with this pathology.

The work is well written, demonstrating experience in the field. The discussion part can be enriched by addressing the answers to the problems below.

-          The paper utilizes and compares a wide range of machine learning classifiers. The study does not explore deep learning models like CNNs or LSTMs, were such types of models considered, and if not, for what reasons?

-          The use of advanced optimization frameworks like Optuna, RayTune, and HyperOpt strengthens the model performance and combined with the large suite of models used, the space of model searched is vast as such this offers high confidence in the shown performance. These tools are known for efficient parameter tuning but can be computationally expensive, it would be interesting to compare the results against simpler methods (e.g., grid or random search), to offer a better understanding of the compute/performance trade-off these approaches offer.

-          SHAP values were used to explain the model’s predictions, which is a valuable addition for transparency in a clinical setting to improve confidence in results. However, the insights from SHAP don't seem to have been used to adjust the models, change the “ranking” of the models proposed, improve feature selection, or guide data imputation. Did you consider leveraging SHAP findings to inform feature selection, model tuning, data augmentation etc? If not, what prevented you from doing so?

-          The dataset of 72,961 colorectal cancer patients is a notable strength and allows for robust model training. Since the dataset is specific to Brazilian patients, do you consider the model would generalize well on other populations? Are there any specific biases that would need to be considered?

-          The KNN-based approach for handling missing data might not be the most appropriate method, were other imputation techniques that introduce less bias considered? (such as EM or Bayesian methods)         

However, self-citation of references 16 and 17 does not seem particularly necessary for the economy of the article.

Reviewer 2 Report

Comments and Suggestions for Authors

Briefly, the authors are providing research on a highly important subject of AI and machine learning in biomedical research with a potential translatory effect. The subject of their study is development of effective classification models in colorectal cancer that would predict the survival of CRC patients in order to help apply appropriate therapy treatments.

They compare three frameworks for parameters optimization for the first time: Optuna, RayTune and HyperOpt.

The database used, methods and results were well described and scientifically solid.

I have several minor comments and suggestions:

1. Fig.1 and Fig. 4 should have bigger font on their axes. Fig. 4 especially is not readable.

2. The authors should include a table with number of deaths/survivals for each category (1 year, 3 years, 5 years), either in the methods section or the results section. In their presentation the reader is not aware of numbers, and in my opinion it would increase the impact of the results. We would exactly know the number in each category that the models were trained and tested on. This is an important point, as it may contribute to explaining the real weight of the model, or point to weaknessis in its development. It may also have an impact on the procedure of data collection in the clinical settings, in the sense that it may serve as a guideline to obtain better dataset for future machine learning, which is for sure a tool of tomorrow in personalized medicine.

3. In the discussion the authors mention that they plan to include PCA analysis in the future. Although this is publishable as is, in my opinion the paper would benefit from inclusion of PCA analysis at this stage.
